# Pneumococcal vaccination at 65 years and vaccination coverage in at-risk adults: A retrospective population-based study in France

**Benjamin Wyplosz[1]\*, Benjamin Grenier [2], Nicolas Roche[3], François Roubille[4], Paul Loubet[5], Ariane Sultan[6], Bertrand Fougère[7,8], Jérôme Fernandes[9], Didier Duhot[10], Bruno Moulin[11], Fanny Raguideau[2], Emmanuelle Blanc[12], Gwenael Goussiaume[12]**

**1** AP-HP, Hospitalisation à Domicile, Département Adulte, Institut Pasteur, Centre Médical, Paris, France, **2** Epidemiology Department, Heva, Lyon, France, **3** AP-HP, Pneumology, Cochin Hospital, Paris, France, **4** PhyMedExp, University Montpellier, CNRS, INSERM, Cardiology, Montpellier University Hospital, Montpellier, France, **5** Virulence Bactérienne et Infections Chroniques, INSERM U1047, University Montpellier, Service des Maladies Infectieuses et Tropicales, CHU Nîmes, Nîmes, France, **6** PhyMedExp, University Montpellier, CNRS, INSERM, Endocrinology-Diabetology-Nutrition Department, University Montpellier, Montpellier, France, **7** Division of Geriatric Medicine, Tours University Hospital, Tours, France, **8** EA 7505 Education, Ethics, Health, Tours University, Tours, France, **9** Medical Information Department, Bayonne Hospital, Bayonne, France, **10** Société Française de Médecine Générale, Issy les Moulineaux, France, **11** Nephrology and Transplantation, Strasbourg University Hospital, Strasbourg, France, **12** Pfizer Vaccines, Paris, France

\* benjamin.wyplosz@aphp.fr

## Abstract

### Objectives

Age (> 50 years) is a risk factor for pneumococcal disease, but is not an indication for vaccination in France, by contrast to influenza. In 2018, the pneumococcal vaccine coverage rate (VCR) was 4.5% in adults at-risk, in contrast to the influenza VCR, which was 43.6%. We aimed to assess pneumococcal and influenza VCR in 2020 in the entire French population and factors associated with a higher VCR (including the age of 65 years).

### Methods

We retrospectively included all adults covered by the National Health Service in 2020 and identified patients at-risk using validated algorithms. We assessed VCRs by analysing pneumococcal vaccines reimbursed between 2009 and 2020 (13-valent pneumococcal conjugate vaccine [PCV13] and 23-valent polysaccharide vaccine [PPSV23]), and influenza vaccines reimbursed between September 2020 and March 2021.

### Results

In 2020, we identified 7,336,769 adults at risk (median age: 67.0 years): 84.2% had comorbidities and 24.5% were immunocompromised. The overall pneumococcal VCR

**Data availability statement:** Restrictions apply to the availability of the data supporting the study findings as they contain potentially identifying and sensitive patient information. These data cannot be shared publicly as they are part of the National health data system (SNDS, Système national des Données de Santé). They are available from the HDH (Health Data Hub https://www.health-data-hub.fr/). Special permission to access these data for this study was granted by the Ethics and scientific committee for health research, studies, and evaluations (CESREES, Comité Ethique et Scientifique pour les Recherches, les Etudes et les Evaluations dans le domaine de la Santé) (former CEREES, file No. 3405058) and the French data protection authority (Comité National de l'Informatique et des Libertés, CNIL, file No. 921156, decision DR-2021-096).

**Funding:** The financial support for this study was provided by Pfizer. The funders played no part in the study's design, data collection and analysis, or the decision to publish. GG and EB, who are employed by Pfizer, provided assistance with the preparation of the manuscript by offering critical feedback on the content prior to its submission. The funder provided support in the form of salaries for authors BW, NR, FR, PL, AS, BF, JF, DD, BM, but did not have any additional role in the study design, data collection and analysis, or decision to publish. The specific roles of these authors are articulated in the 'author contributions' section.

**Competing interests:** I have read the journal's policy and the authors of this manuscript have the following competing interests: BW received Payments from for lectures from Pfizer, Sanofi, Lilly, GSK, for meetings from Pfizer. BG and FR are employees of Heva, the CRO which received payments from Pfizer to conduct the study. NR received institutional grants, consulting fees, honoraria for lectures from Chiesi, GSK, Pfizer, consulting fees and honoraria for lectures from AstraZeneca, Sanofi, Teva, consulting fees from Austral and Biosency, honoraria for lectures, from Zambon, MSD, Menarini and support for meetings from Chiesi, AstraZeneca, GSK. FRoub received honoraria for lectures from Astra Zeneca, Servier, Boehringer, Astra Zeneca, Vifor, Bayer, Pfizer, Novartis, Servier, Novonordisk, Air liquid, Abbott, QuidelOrtho,

(PCV13 + PPSV23) was 9.9% and the seasonal influenza VCR was 51.1%. The variable associated with the highest odds of VCR was an "age ≥65 years" for influenza (odds ratio [OR] 4.14), but not for pneumococcal vaccination (OR 1.02). In patients with comorbidities, pneumococcal VCR did not significantly increase between those aged 18–65 years and those aged > 65 years (7.2% to 9.4%), and even decreased from 20% to 17.9% in patients with immunodeficiencies. In contrast, influenza VCR increased significantly from 35.5% to 67.9% (OR 3.55) in patients with comorbidities, and from 27.3% to 71.2% (OR 5.57) in those with immunodeficiencies.

## Conclusion

In France, pneumococcal VCR did not increase above 65 years of age (OR 1.02), by contrast to influenza VCR (OR 4.14) that increased significantly, suggesting that an age-based recommendation for pneumococcal vaccination will probably benefit to VCR in at-risk, elderly population.

## Introduction

Age (> 50 years) is an independent risk factor for infections, particularly for influenza and pneumococcal disease [1]. In Europe, the incidence of invasive pneumococcal community-acquired pneumonia (CAP) in people aged ≥65 years has been estimated to be between 16 and 95 cases per 100 000 in different countries, with an incidence increasing with age [2]. In the USA, the incidence of invasive pneumococcal disease (IPD) was recently estimated to be 4.1/100 000 in healthy adults aged 50−64 years compared with 9.8/100 000 in those aged 65 years and older [3]. In France, pneumococcal pneumonia is more frequent, its incidence in the 65 years and above population has been recently estimated to range from 34.7 to 130/100000, depending on the type of associated underlying comorbidities [4,5]. However, the incidence of IPD is also a dynamic process that evolves in response to the use of pneumococcal conjugate vaccines. The PSERENADE (Pneumococcal Serotype Replacement and Distribution Estimation) project showed that 30 European countries saw a rapid reduction in the incidence of total IPD in young children, reaching 58−74% in the 6 years following the introduction of PCV10 or PCV13 into their routine vaccination schedules [6]. This effect was followed by declines in older children due to indirect effects, and more moderately in adults by 25% (95% CI 21–36) to 29% (20–30) in PCV-13 using countries. In France, PCV-13 was introduced in 2010. After a 5-year decrease in the incidence of IPD in adults, the pneumococcal serotype replacement led to a rebound in cases that reached, in 2017, to the pre PCV-13 level [5]. In 2024, the French health authorities simplified the pneumococcal vaccination schedule (PCV13 + PPSV23) to a single dose of PCV20 for adults [7].

According to the ECDC, 21 (70%) of the 30 European Economic Area (EEA) countries have implemented age-based pneumococcal vaccination strategies for adults (60 or 65 years). The remaining countries (30%, N = 9) have either no recommendations for adult pneumococcal vaccination (N = 7) or recommendations based

Newcard, MSD, BMS, Sanofi, Alnylam, Zoll, Implicity, GSK, BMS, consulting fees from Abbott, Air liquide Bayer, Pfizer, support for travel from Novartis, Boehringer-Ingelheim, participates in an advisory board for Carmat, fiduciary role for Boehringer-Ingelheim, Vifor Pharma, Novartis, medical writting from Pfizer. PL received consulting fees from Pfizer, GSK, Sanofi, honoraria for lecture from Pfizer, GSK, Sanofi, Moderna, MSD, Support for attending meeting from Pfizer, Astrazeneca, MSD, Sanofi, GSK. AS received Grants from VIATRIS, SERVIER, Consulting fees from PFIZER, SANOFI, LILLY, SERVIER, ASTRA, DEBEX, NOVARTIS, URGO, honoraria for lectures from PFIZER, SANOFI, LILLY, SERVIER, ASTRA, DEBEX, NOVARTIS, ABOTT, MSD. BF received Consulting fees from Sanofi, AstraZeneca, Pfizer, GSK, honoraria for lectures from Sanofi, AstraZeneca, Pfizer, GSK, CSL Seqirus, Participation on a Data Safety Monitoring Board or Advisory Board for Pfizer. JF received Grants from Astra ZENECA ASTELLAS Pharma ABBVIE JANSSEN BMS, Consulting fees from Astra ZENECA ASTELLAS Pharma ABBVIE JANSSEN BMS, honoraria for lectures from Astra ZENECA ASTELLAS Pharma ABBVIE JANSSEN BMS. DD received honoraria for lecture from Pfizer, BM received honoraria for lecture from Pfizer. EB and GG are employees from Pfizer and may hold stock options. This does not alter our adherence to PLOS ONE policies on sharing data and materials.

on risk factors (N = 2), including France [8,9]. Strategies based on age rely on the concept of "healthy ageing", which is easy for both patients and doctors to understand and for health authorities to implement at a national level [10]. It is considered to be one of the most effective interventions for increasing vaccination coverage in the general population [11,12]. The strategy based on specific comorbidities and risk factors is "patient-centred" and relies on the expertise of health professionals to identify patients at risk. It requires clinicians to be aware of all chronic conditions that increase the risk of pneumococcal disease and is sensitive to their commitment to prescribe vaccines. As it does not take into account age as an independent risk factor for pneumococcal disease, it misses all healthy elderly people who are at risk of pneumococcal disease only because of their age (about 13.5 million people aged ≥ 65 years were living in France in 2020, but the proportion of those living with ≥ 1 risk factor is not known) [13].

France and Finland are the two countries in the EEA that have implemented a pneumococcal vaccination policy based on immunisation of at-risk patients only [5]. Although both vaccinations target roughly the same population, their schedules are quite different. For influenza vaccination, the National Health Service sends an annual prescription to adults at risk based on both age (≥65 years) and the presence of a chronic disease, whereas in the absence of an age-based recommendation like for pneumococcal vaccination, physicians have to individually identify eligible at-risk adults. In a previous study, in 2018, we identified 4 million at risk adults among 75% of the French adult population (N = 49,232,522) [14]. Pneumococcal VCR (combined regimen with a 13-valent pneumococcal conjugate vaccine [PCV13] and a 23-valent pneumococcal polysaccharide vaccine [PPSV23] [15] was estimated to be at about 4.5% and influenza VCR (one dose of annual seasonal vaccine) at about 43.6%. We showed that pneumococcal VCR did not increase significantly with the number of general practitioner (GP) or specialist consultations, suggesting difficulties for doctors in identifying adults at risk of pneumococcal disease and a need for raising vaccine education and awareness among health care professionals.

An even simpler way to increase pneumococcal VCR will be to set the age of eligibility for pneumococcal vaccination at 65 years, for both healthy and at-risk adults, as it is already the case for influenza in France. To assess the potential impact of this measure, we used the French nationwide claims database (SNDS), with algorithms covering approximately 100% of at-risk patients, to estimate pneumococcal and influenza vaccination coverage in at-risk patients in 2020. We then compared the factors associated with a better VCR, focusing on one hand on the age of 65 years, and on the other hand on the identification of risk factors.

## Materials and methods

### Study design and data sources

This is a retrospective observational study performed on claims data from the National Health Data System (SNDS) to determine vaccination coverage in at-risk adults. This claims database covers 66.8 million people (98.8%) out of the French population. It contains demographic data (dates of birth and death, sex, place of

residence), and medical information (drugs, medical devices, medical procedures, laboratory analyses, hospitalizations with diagnosis-related groups coded according to the International Classification of Diseases 10th Revision [ICD-10]) [16–20].

## Study population

We included all adult patients (≥18 years old) with a unique identifier, who were affiliated to the General National Health Insurance (NHI) scheme between 1 January and 31 December 2020 (1 September 2020 and 31 March 2021 for influenza). We then identified patients at risk of pneumococcal disease, following the list of medical conditions provided by the Haut Conseil de la Santé Publique (HCSP) in its 2017 recommendations [21].

In brief, immunocompromised conditions at risk of pneumococcal disease were: asplenia or hyposplenia, hereditary immunodeficiency, HIV infection, solid organ transplant (SOT) candidates and recipients, chronic autoimmune or inflammatory disease treated with immunosuppressive or biologic drugs, nephrotic syndrome, or were treated with chemotherapy for a solid tumour or a hematologic malignancy. Stem cell transplant recipients were not included in the study because they follow a different vaccination schedule (three doses of PCV13 and one dose of PPV23). Comorbidities at risk of pneumococcal infection were: chronic respiratory disease (including chronic obstructive pulmonary disease [COPD], asthma, cystic fibrosis, emphysema, lung cancer, interstitial lung disease [22,23]), chronic heart disease, end-stage renal failure with replacement therapy, chronic liver disease, diabetes requiring treatment, an osteo-meningeal breach, or a cochlear implant. Patients with multiple conditions were counted once for each condition but only once in the grand total. Algorithms are described in S1 Methods. Four algorithms were used by French Health authorities for the surveillance of HIV, end-stage renal failure with replacement therapy, chronic liver disease, and diabetes requiring treatment [24]; three were modified for our study (patients treated for cancer, auto-immune diseases, chronic respiratory diseases); the one for severe asthma was already published [25], and seven were created for our study (asplenia, hereditary immune deficit, solid organ transplantation, cyanotic cardiac disease, osteo-meningeal breach, and cochlear implant). We did not formally validate those latter, but we assessed the external validity by comparing the figures with available external data [26–28].

## Vaccine tracking

At the time of the study, the recommended pneumococcal vaccination schedule consisted of administering of a 13-valent pneumococcal conjugate vaccine (PCV13) followed by a 23-valent pneumococcal polysaccharide vaccine (PPSV23) between two and twelve months later [15]. We tracked PCV13 and PPV23 vaccines dispensed in community pharmacies in the database. The outcome measure was the pneumococcal vaccination coverage (corresponding to the reimbursement of one dose of PCV13 and one dose of PPV23 less than one year later) in the population at risk of pneumococcal disease.

We defined influenza vaccination coverage as a dose dispensed between 1 September 2020 and 31 March 2021. Influenza vaccination coverage was also determined in patients at risk of pneumococcal disease.

## Health care visits

All health care visits are recorded in the SNDS. The number of hospital admissions, nurse visits, GP visits and specialist visits per patient were reported.

## Data analysis

As the SNDS covers almost the entire French population, no power calculation, sample size calculation or imputation of missing values was required [20].

Continuous data were summarized as mean with standard deviation (SD) or median with first and third quartiles (Q1 and Q3). Categorical data were summarized as numbers and percentages.

Vaccination coverage was assessed overall and for each subgroup of interest, as well as for the following age groups: 18–65 years, and > 65 years. It was assessed for the pneumococcal vaccination coverage, and for at least one PCV13 dispensing, and for at least one PCV13 and/or PPSV23 dispensing.

If the level of significance (p-value) is almost always "statistically significant" and the power of the study is acceptable (here the sample size is a whole population), it is necessary to assess whether "statistically significant" factors are "clinically significant" (the intervention may have influenced the vaccine uptake), we relied on the value of the odds ratio (OR) as an effect size index that is independent of the sample size, considering that an OR = 1.5 corresponds to a small effect size, an OR = 2 to a medium effect size and an OR = 3 to a large effect size [29].

In a multivariable logistic regression model, factors associated with either pneumococcal or influenza vaccination were analysed separately. Included variables were: age, sex, number of comorbidities or immunodeficiencies, number of hospital admissions, and visits to GP, private specialist, and community nurses.

Statistical analyses were performed using SAS software, version 9.4 (SAS Institute Inc., Cary, NC, USA).

### Ethical approval and consent to participate

Authorization to use the data was granted by the French data protection authority (*Commission Nationale de l'Informatique et des Libertés*, CNIL) (Decision DR-2021–096, and authorization No. 921156) and the study protocol was approved by the ethics and scientific committee for research, studies and evaluations in the field of health (*Comité éthique et scientifique pour les recherches, les études et les évaluations dans le domaine de la santé*, CESREES) (Dossier 3405058, on 18 March 2021). In accordance with the current regulations, patient consent was not required because the study used secondary data and because of the public interest in the evaluation and because the protection of patients' rights and freedoms was guaranteed. Data were accessed from May 6 2022 until November 30 2024. The database uses unique, anonymous identifiers transformed from social security numbers of patients, allowing for the data capture of individual-level healthcare claims of over 99% of the French population. No sensitive variables posing a risk of patient reidentification had been accessed for this study.

## Results

### Study participants characteristics

In 2020, we identified 7,336,769 adults (48.2% of whom were women) covered by the French National Health Insurance scheme with at least one chronic condition that puts them at risk of pneumococcal disease, according to French vaccination guidelines (Fig 1). Of these, 6,175,172 (84.2%) had at least one comorbidity, and 1,796,392 (24.5%) had at least one immunosuppressive condition. Patients could belong to both categories, so that the sum of immunocompromised and comorbid patients exceeded the total number of patients. In total, 27.6% of the cohort (N = 2,026,863) had two or more chronic conditions at risk of pneumococcal disease.

The five most common comorbidities were: diabetes (N = 3,825,602; 62.0%), chronic respiratory disease (N = 1,991,052; 32.2%), chronic heart disease (N = 864,776; 14.0%), chronic liver disease (N = 525,973; 8.5%), and end-stage renal failure on replacement therapy (N = 83,873; 1.4%). The most common immunosuppressive conditions were chemotherapy-treated malignancies (N = 835,107; 46.5%), autoimmune or inflammatory diseases (N = 604,033; 33.6%), nephrotic syndrome (N = 191,962; 10.7%), and HIV infection (N = 166,190; 9.3%) (Fig 1).

Table 1 shows the characteristics of the study population. The mean age (±SD) was 65.0 (±15.6) years; 3,881,151 (52.9%) were older than 65 years. Patients with comorbidities were older than those with immunosuppression (66.6 ± 14.6 versus 59.8 ± 17.5), and most of them (56.7%) were older than 65 years.

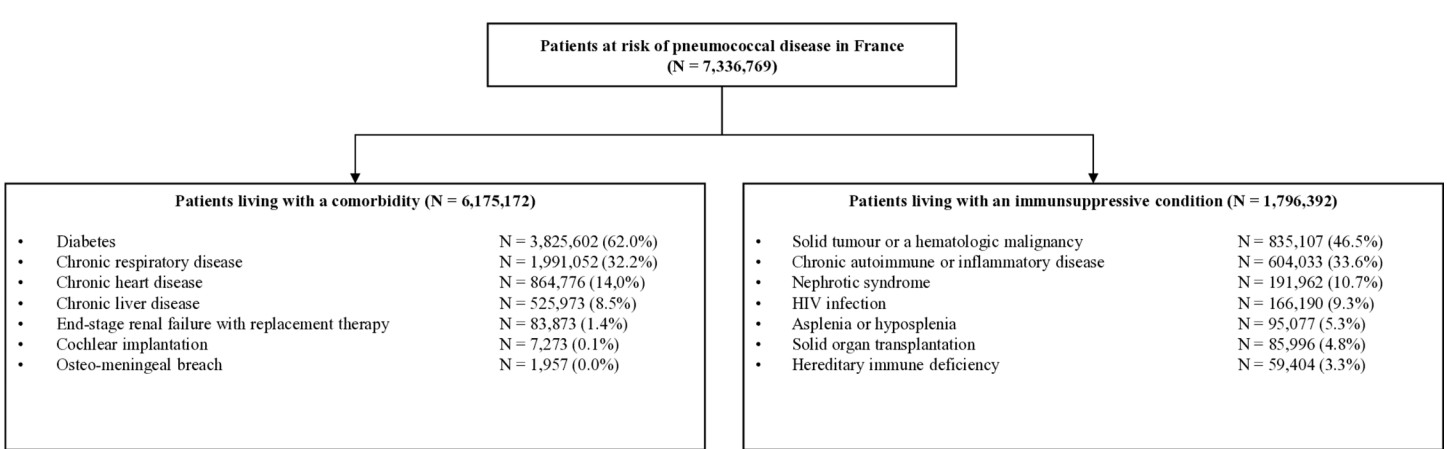

**Fig 1. Study population: patients living with comorbidities (on the left) and immunosuppressive conditions (on the right) who are at risk of pneumococcal disease in France in the year 2020.** Patients could be classified in both categories, so the total number of patients living with a comorbidity and those living with an immunosuppressive condition exceeded the total number of patients.

**Table 1. Characteristics of patients at risk of pneumococcal disease in France in 2020 and vaccination rates.**

| Population | Comorbidities | Immunodepression | Total |
|---|---|---|---|
| Patients, N (%) | 6,175,172 (84.2) | 1,796,392 (24.5) | 7,336,769 (100.0) |
| Women, N (%) | 2,827,988 (45.8) | 1,005,28 (56.0) | 3,538,000 (48.2) |
| Age | | | |
| Mean (SD) | 66.6 (14.6) | 59.8 (17.5) | 65.0 (15.6) |
| Median (Q1-Q3) | 68.0 (58-77) | 61.0 (47-73) | 67.0 (55-76) |
| Age groups, N (%) | | | |
| 18–65 years | 2,676,628 (43.3) | 1,046,248 (58.2) | 3,455,847 (47.1) |
| >65 years | 3,498,544 (56.7) | 750,144 (41.8) | 3,880,922 (52.9) |
| Visits to a health professional in 2020 | | | |
| General practitioner office visits | | | |
| Patients with ≥ 1 visit, N (%) | 5,345,506 (86.6) | 1,514,578 (84.3) | 6,322,882 (86.2) |
| Visits by patient, mean (SD) | 5.9 (5.9) | 6.1 (5.6) | 6.0 (5.6) |
| Private specialist physician office visits | | | |
| Patients with ≥ 1 visit, N (%) | 2,976,109 (48.2) | 1,034,606 (57.6) | 3,650,105 (49.8) |
| Visits by patient, median (IQR) | 2.7 (8.4) | 1.7 (6.2) | 1.8 (6.0) |
| Community nurse visit | | | |
| Patients with ≥ 1 visit, N (%) | 4,738,226 (76.7) | 1,447,679 (80.6) | 5,638,473 (76.9) |
| Visits by patient, median (IQR) | 30.3 (74.2) | 31.7 (85.5) | 29.4 (81.1) |
| Hospital admissions in 2020 | | | |
| Patients with ≥1 admission, N (%) | 2,071,676 (33.6) | 955,484 (53.2) | 2,608,480 (35.6) |
| Admissions by patient, median (IQR) | 4.7 (14.7) | 1.3 (7.8) | 1.6 (7.6) |
| Vaccination coverage (%) | | | |
| Pneumococcal vaccination | | | |
| •Pneumococcal vaccination coverage | 8.5 | 19.1 | 9.9 |
| •At least one PCV13 dispensing | 11.7 | 25.6 | 13.6 |
| •At least one PCV13 and/or PPSV23 dispensing | 19.7 | 35.2 | 21.7 |
| 2020−21 seasonal influenza vaccination | 53.8 | 45.4 | 51.1 |

SD: Standard deviation. Patients can be identified in both categories, leading to a sum of immunocompromised patients and patients with a chronic condition > total of patients.

To determine whether a possible low vaccination rate could be explained by a lack of contact with the healthcare system, we described the number of visits to a GP, specialist, or nurse in 2020 and the number of hospital admissions. Table 1 shows that more than 86% of patients at risk visited a GP at least once in 2020, about 50% visited a private specialist and more than 76% visited a nurse. Hospital admissions were 33.6% for patients with ≥1 comorbidity, and 53.2% for immunocompromised patients (see S1 and S2 Tables for details by chronic condition). In addition, they all had many contacts with their pharmacist as, in France, long-term medicines are usually supplied for one month.

## Vaccine coverage rates (VCR)

Table 1 shows the VCRs for the whole cohort: pneumococcal VCR (PCV13 + PPV23) was 9.9% and influenza VCR was 51.1%. Pneumococcal VCR was 19.1% in immunocompromised patients but 8.5% in patients living with ≥1 comorbidity (Fig 2A and 2B). More sensitive definitions of the vaccination coverage (at least one PCV13, at least one PCV13 and/or one PPSV23) reported higher rates, regardless of being immunocompromised or living with a comorbidity (Table 1). Conversely influenza VCR was 53.8% in patients living with a comorbidity and 45.4% in those living with an immunosuppression condition. Further details by disease and by region are shown in S1 File.

**Pneumococcal vaccination (Fig 2).** Overall, the estimated pneumococcal VCR was 8.5% for patients with a comorbidity, ranging from 5.6% (diabetes) to 37.2% (cochlear implant). Patients living with diabetes, which was the most prevalent cohort (62.0%), had the lowest VCRs (5.6%), while patients living with an end-stage renal disease or a chronic respiratory disease had some of the highest VCRs, (33.3% and 16.2%, respectively).

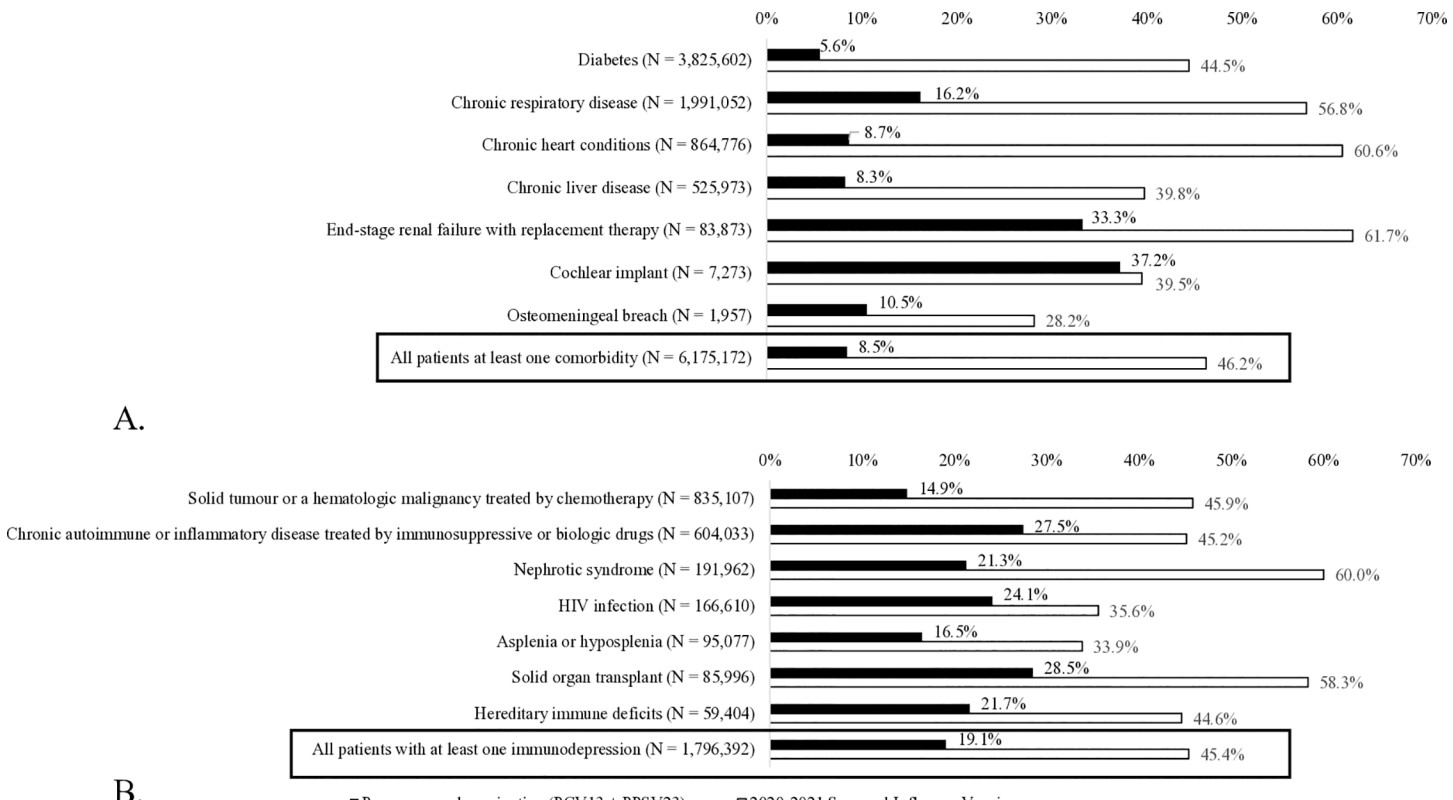

**Fig 2. Pneumococcal (black bars) and influenza (white bars) vaccine coverage in patients living with comorbidities (A), and immunosuppressive conditions (B), who are at risk of pneumococcal disease in France in 2020.** Populations are listed in descending order of numerical value.

The estimated pneumococcal VCR of patients living with immunosuppression ranged from 14.9% (patients treated for a malignancy) to 28.5% (solid organ transplant recipients) (Fig 2B). The most prevalent group of immunocompromised persons (patients treated for a solid tumour or haematologic malignancy) had the lowest pneumococcal VCR (14.9%). Only 16.5% of asplenic persons were vaccinated against pneumococcal infections.

**Influenza vaccination (Fig 2).** Estimated influenza vaccination rates in patients with comorbidities were much higher than the pneumococcal vaccination rates, ranging from 28.2% for those with a osteomeningeal breach to 61.7% in those with end stage renal failure with replacement therapy. Patients with chronic respiratory or heart disease also had a vaccine coverage of over 50%.

The same observation was made for influenza vaccination in immunocompromised patients, and to a similar extent: from 33.9% in people with hypo- or asplenia to 60% in those with nephrotic syndrome. The other group of patients with a vaccine coverage rate over 50% were transplant patients.

### VCR by age groups and factors associated with vaccine dispensing

The overall pneumococcal VCR was similar in patients aged 18–65 years (9.8%) and in those aged 65 years and older (10.0%; OR 1.02). It increased slightly in patients with comorbidities, from 7.2% to 9.4% (OR 1.18), respectively (Fig 3A), but decreased significantly with age in patients with immunodeficiencies, from 20.0% to 17.9% (OR 0.75) (Fig 3B).

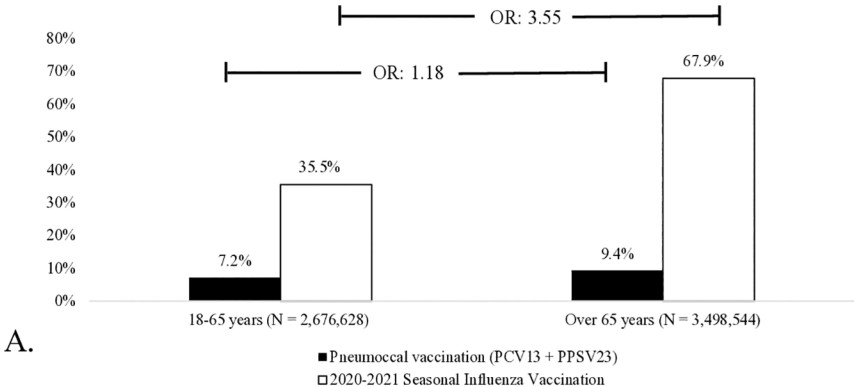

A.

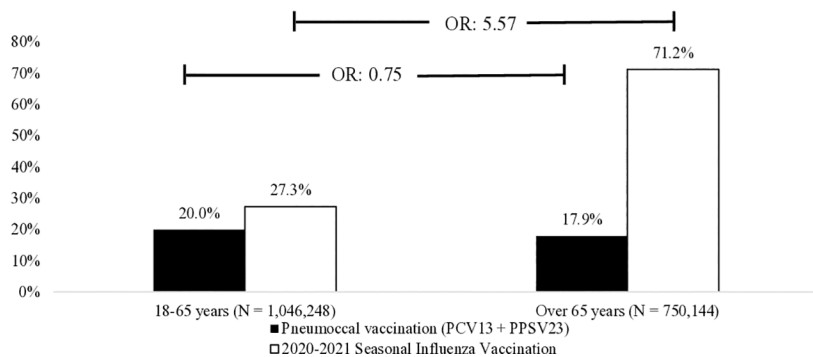

B.

**Fig 3. Pneumococcal (black bars) and influenza (white bars) vaccine coverage in patients living with comorbidities (A), and immunosuppressive conditions (B) according to age of 65 years, in France in 2020.** OR = odds of vaccination coverage in patients ≥ 65 years/odds of vaccination coverage in patients ≤ 65 years. OR = 1.5, small size effect; OR = 2, medium size effect; OR = 3, large size effect.

The overall VCR for influenza was much higher than for pneumococcal vaccination and increased significantly with age: The odds of being vaccinated were 4.14 more important in patients aged 65 years and older (VCR 63.0%) than in those aged 18–65 years (VRC 28.0%). This increase was greater in patients with immunodepression (OR 5.57; 71.2% in those aged > 65 years versus 27.3% in those aged 18–65 years) than in those living with comorbidities (OR 3.55; 67.9% in those aged > 65 years versus 35.5% in those aged 18–65 years) (Fig 3). Details per pathology and per age group are shown in S1a and S1b Figs for pneumococcal vaccination and S2a and S2b Figs for influenza vaccination.

## Patients living with comorbidities

For pneumococcal vaccination, the odds of being vaccinated were 2.26 times [95% CI 2.24–2.27] higher in individuals with two comorbidities than in than those with one comorbidity, after adjusting for other specified variables. Furthermore, the effect was even more pronounced (OR = 3.75 [95% CI 3.71–3.78]) for those with three or more comorbidities compared to those with one comorbidity. In contrast, the odds were smaller in women compared to men (OR 0.98 [0.98; 0.99]) as well as in individuals who have visited their GP 1−6 times in 2020 (OR 0.94 [0.93–0.95]) compared to those with no consultation (Fig 4 and S3 Table).

For influenza vaccination, the odds of being vaccinated were 3.55 times (3.54–3.56) more important in individuals aged > 65 years than in those aged 18−65 years. It was 1.99 times (1.98–2.0) and 2.47 times (2.45–2.48) higher in those visiting a nurse "1-4 times" and "≥ 5 visits" than in those with no visit to a nurse in 2020, after adjusting for other specified variables. The presence of "≥3 comorbidities" (OR 1.49 [1.47–1.50] and "≥7 visits to a GP" (OR 1.52 [1.51–1.53]) had only a small effect (OR ≥ 1.5); whereas being a woman (0.84 [0.83–0.84]) and being "hospitalised (OR 0.89 [0.88-0.89]) 1-4

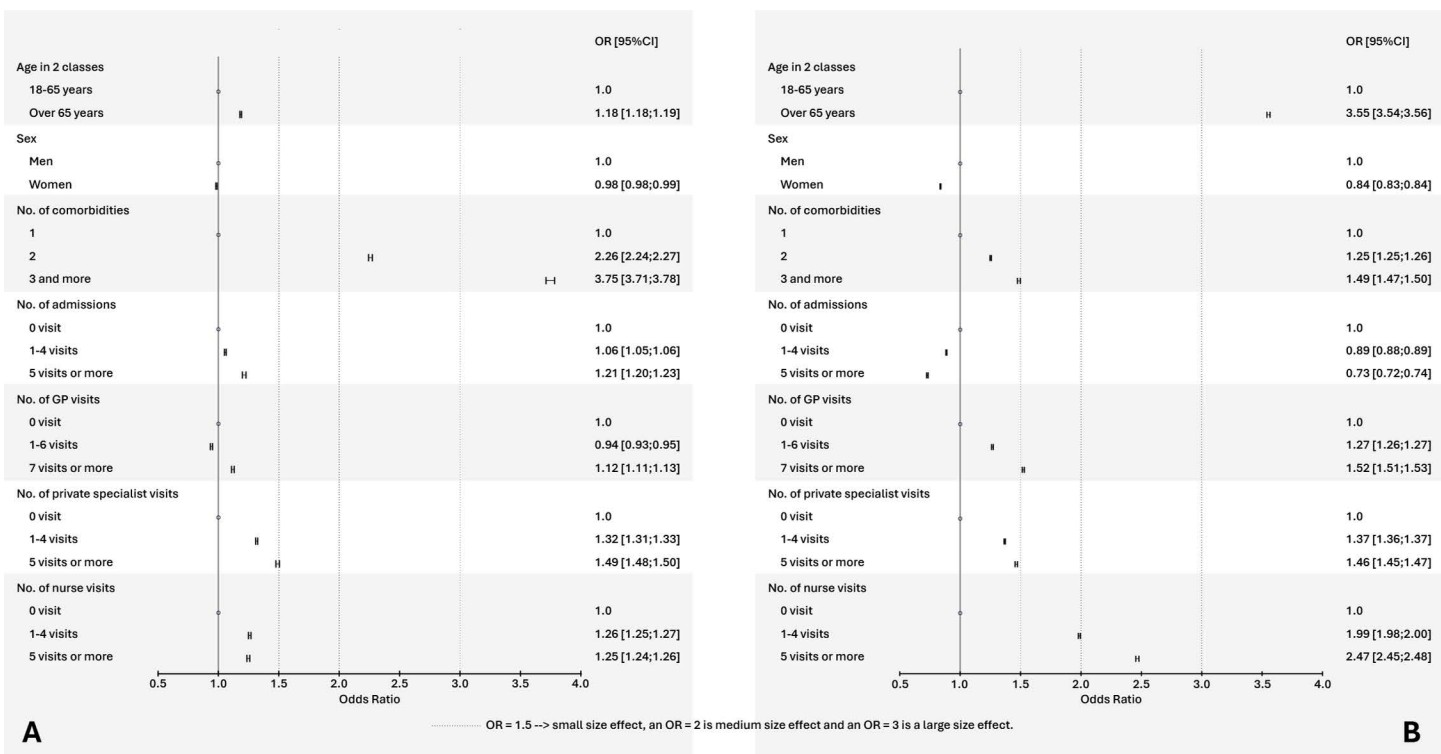

**Fig 4. Characteristics associated with pneumococcal primary vaccination (A) and 2020−21 seasonal influenza vaccination (B) in patients with comorbidities in France in 2020 (multivariable analyses).**

times"; OR 0.73 [0.72–0.74] for "≥5 times in 2020") were associated with smaller odds of being vaccinated against influenza (Fig 4 and S3 Table).

**Patients living with immunosuppressive condition.** For pneumococcal vaccination, the odds of being vaccinated were 2.02 [2.00–2.05] times higher in patients with ≥ 3 comorbidities than in those without comorbidities. All other variables were associated with an OR ≤ 1.5 or even a paradoxical but clearly negative effect size (OR < 1) for pneumococcal vaccination in those aged ≥ 65 years (0.75 [0.75–0.76]), in women (0. 84 [0.83; 0.85]), and in those hospitalised in 2020 (OR 0.86 [0.86–0.87] for "1-4 times"; OR 0.74 [0.73–0.74] for "≥5 times in 2020") (Fig 5 and S3 Table).

For influenza vaccination, four variables were associated with moderate or large odds of being vaccinated: 1) being aged ≥ 65 years (5.47 [5.44–5.51]); 2) having comorbidities (OR 1.58 [1.57–1.60] for "2 comorbidities"; OR 2.08 [2.06; 2.10] for "≥ 3 comorbidities"); 3) visiting a nurse in 2020 (OR 1.75 [1.74–1.77] for "1-4 visits"; OR 2.31 [2.29–2.34] for "≥5 visits"); 4) visiting a GP "≥ 7 times in 2020" (OR 1.52 [1.50–1.53]). Other variables had less than a small effect (OR <1.5), whereas being female (0.74 [0.74–0.75]) or having been hospitalised in 2020 (OR 0.89 [0.88–0.89] for "1-4 times"; OR 0.81 [0.81; 0.82] for "> 5 admissions"), were negatively associated with vaccine dispensation (Fig 5 and S3 Table).

## Discussion

In 2020, we identified 7,336,769 (14.4%) adults at risk of pneumococcal disease (mean age: 65 years ± 15.6) out of the total adult population (51.1 million) covered by the French General National Health Insurance. Of these, 6,175,172 (84.2%) lived with one or more comorbidities (62.0% with diabetes and 32.2% with chronic respiratory disease) and

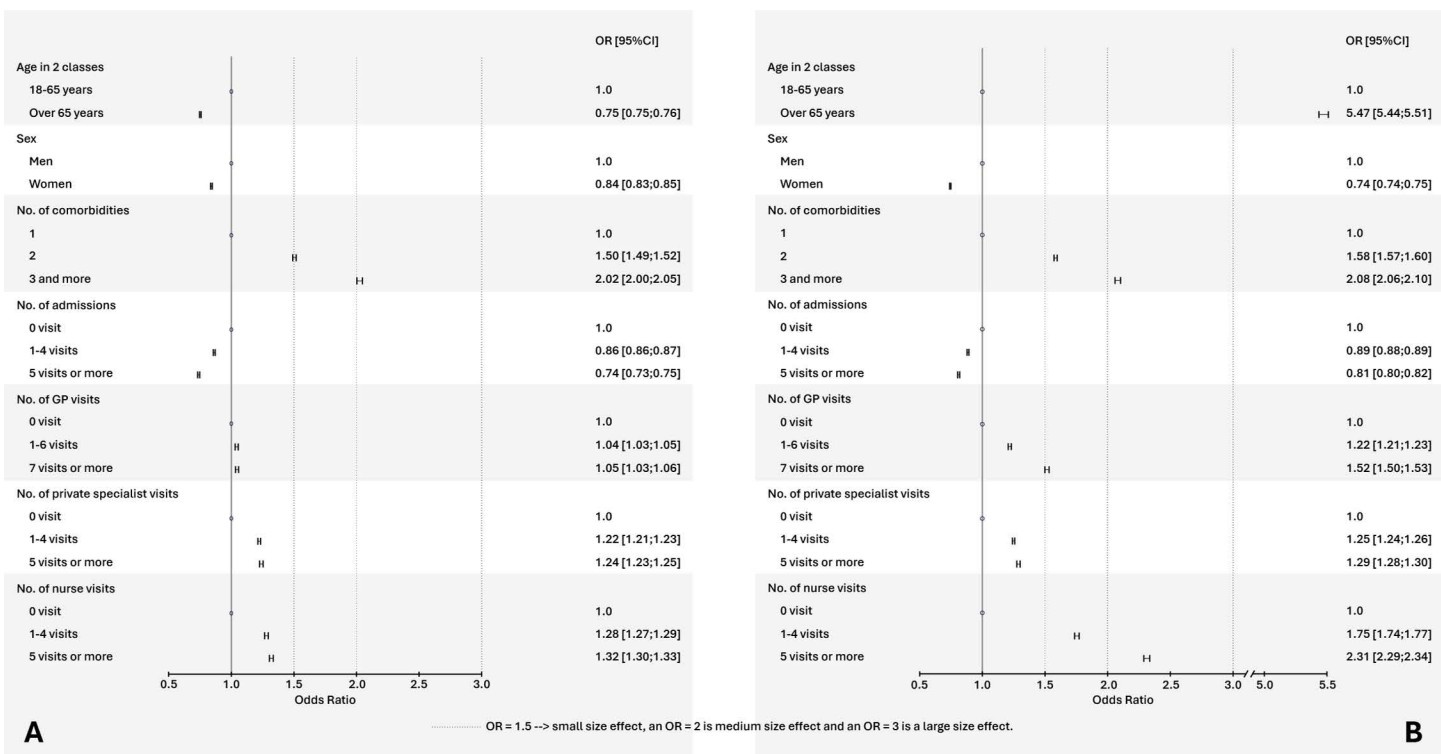

**Fig 5. Characteristics associated with pneumococcal primary vaccination (A) and 2020−21 seasonal influenza vaccination (B) in patients with immunodepression in France in 2020 (multivariable analyses).**

1,796,392 (24.5%) with immunosuppressive conditions (46.5% with chemotherapy-treated malignancies and 33.6% with auto-immune or inflammatory diseases). Despite high frequency of contacts with healthcare professionals (86.2% had seen their GP on average of 6 times during the year), overall vaccination coverage was 9.9% for pneumococcal vaccination and 51.1% for seasonal influenza vaccination, suggesting difficulties for doctors in identifying eligible adults at risk.

The variable associated higher odds of with influenza was an age > 65 years (OR = 4.14), whereas the odds of pneumococcal coverage was smaller in those aged > 65 years (OR = 1.02). Although an age ≥ 65 years is an independent risk factor for many respiratory infections, pneumococcal vaccination was neither recommended nor reimbursed in France for adults without any other risk factors, which may explain this lack of vaccination coverage after 65 years. Influenza vaccination, on the other hand, benefits from an intensive national policy, with an annual prescription voucher sent at home by the national health system to eligible at risk and ≥ 65 years patients. Despite the COVID pandemic in 2020, the public health objectives for influenza vaccination (75%) were nearly achieved in the 65 + age group among those with comorbidities (67.9% versus 35.5% in those <65 years; OR 3.55) and those with immunodeficiency conditions (71.2% versus 27.3%; OR 5.57), suggesting that an age-based recommendation is effective in increasing the vaccine uptake.

The results for pneumococcal vaccination in our study were more mixed than for influenza vaccination, with slightly higher vaccination coverage in those with comorbidities (9.4% versus 7.2%; p < 0.001) than in those with immunodeficiencies (17.9% versus 20.0%; p < 0.01). In contrast, in the US, where age 65 years is recognised as an indication for pneumococcal vaccination, vaccination coverage in 2020 was 63.6% in the 65–74 age group and 73.2% in the 75 + age group, but 18.5% in the 45–64 age group. In the United Kingdom, where pneumococcal vaccination is also recommended for those 65+ (one dose of PPSV-23 vaccine), coverage was 70.6% in the 65 + age group, compared with a range of 39% to 56% in the 18–64 age groups with various chronic conditions (excluding cochlear implant recipients, who had 70.9% coverage). [1] These results suggest that an age-based recommendation for pneumococcal vaccination is likely to improve the coverage.

For influenza vaccination, patients who saw a nurse in 2020 presented higher odds of being vaccinated than those who did not (OR 1.99 for "1-4 visits" for those with comorbidity and OR of 1.75 for "1-4 visits" for those with immunodeficiency), possibly because of the role of nurses in advising and administering the vaccine to at risk patients, while the variable "≥ 3 comorbidities" (OR 2.08) was also associated with higher odds of coverage in the latter group.

For pneumococcal vaccination, the presence of comorbidities, which was the indication of pneumococcal vaccination in France at the time of our study, was associated with a better coverage for both patients with comorbidities (OR 2.26 for "2 comorbidities"; OR 3.75 for "≥3 comorbidities") and in patients with immunodeficiency (OR 2.02 for "≥3 comorbidities"). A small, but negative, effect size was observed in women for those with both comorbidities (OR 0.98) and immunodeficiency (OR 0.84), which may be related to higher vaccine hesitancy in women [30,31]. Other variables associated with a lower odds of coverage were: "1-6 visits to a GP in 2020" (OR 0.94) in the former group, possibly because of the impact of the COVID pandemic on the health system; and having been hospitalised in 2020 (OR 0.86 for "1-4 admissions" and OR 0.74 for "5 or more admissions") in the latter group. We could not to determine why adults hospitalised in 2020 were less likely to be vaccinated, but it could reasonably be explained by the COVID crisis and the lack of reimbursement for pneumococcal vaccines in healthcare facilities in France. It is noteworthy that the number of medical consultations, both with a GP and a specialist, was not associated with a significant effect size (OR 1.5) and was even negative for patients with comorbidities (OR 0.94 in "1–6 GP consultations). We observed similar results in our first study: specialists prescribed an average of one pneumococcal vaccine and GPs prescribed an average of nine vaccines during the 9-year study period [11].

Limitations of our study were that we assumed that reimbursed vaccines were administered, which may have overestimated vaccination coverage. Doses of vaccine may have been forgotten at the pharmacy or brought home but poorly stored at a temperature above +4°C, or even incorrectly administered (a too short or a too long delay between the two vaccines; both vaccines administered on the same day; or the PPV23 administered before the PCV13). On the contrary, some patients may have been vaccinated with free vaccines in a hospital or dispensary, and therefore not included in our

study, but this type of circumstance was probably exceptional. Another limitation may have been that our study took place during the COVID pandemic, which disrupted the healthcare system in 2020. However, we observed vaccination coverage for influenza and pneumococcus that was higher than in 2018 [14] and similar figures have been observed in other countries, suggesting that COVID did not have a deleterious effect on these 2 vaccinations in adults at risk [32,33].

The main strengths of our study were that it was based on real-life data from a nationwide database, covering 98.8% of the French adult population, and that our participant detection algorithms were validated by a previous study and refined over time by the national health system with dedicated expert physicians [21,24]. Thus, the pneumococcal vaccination coverage was higher (9.9%) but more accurate than in our previous publication (4.5%), which was based on only 76% of the adult population in the years 2014–2018 [14].

Our results suggest that a 65 years age-based recommendation for pneumococcal vaccination will help to achieve public health goals for vaccination coverage, as it is the case for influenza vaccination. This strategy has already been adopted by the French health authorities for the vaccination against tetanus, shingles, RSV and COVID. Such a measure is very easy to implement, but will not be sufficient on its own, given that even in the population over 65, the VCRs are below the 75% of vaccine coverage recommended by WHO. The population under 65 is also largely under-vaccinated against either influenza or pneumococcal disease. Other measures to increase vaccination coverage could also be proposed, but most are costly and take time to implement: 1) improving vaccine education for health professionals [34]; 2) expanding the range of vaccine prescribers to include pharmacists, nurses, and midwives [35,36]; 3) developing incentive-based vaccination policies including pay-for-performance in primary care; 4) promoting the use of a single-dose 20-valent pneumococcal conjugate vaccine regimen [7,37,38]; 5) incorporating vaccination into care pathways (vaccination at diagnosis of chronic disease or initiation of specific treatments); and 6) integrating vaccination coverage and recommendations into electronic medical records for healthcare professionals as a reminder of their patients' vaccination status and requirement. Finally, sending pneumococcal vaccine vouchers to at risk patients (about 10 million adults aged over 65 years of age and about 3.5 million at risk of pneumococcal disease under 65 years of age) may be one of the best solutions to consider, as has been seen with the influenza vaccine, which is well accepted despite being given every year [39].

To conclude, the results presented here confirm that pneumococcal vaccination coverage in the French at-risk population is markedly insufficient at the time of our study, despite recommendations to provide vaccination to patients with comorbidities and immunosuppressive conditions predisposing to invasive pneumococcal disease. These results suggest that setting the age of 65 as the threshold for pneumococcal vaccination could be a very simple and cost-effective way to increase vaccination coverage.

## Supporting information

**S1 Methods.  Algorithms to identify the cohorts.**
(DOCX)

**S1 Table.  Characteristics of the patients with a comorbidity associated with an increased risk of pneumococcal disease in France in 2020.**
(DOCX)

**S2 Table.  Characteristics of the immunocompromised patients in France in 2020.**
(DOCX)

**S3 Table.  Characteristics associated with pneumococcal primary vaccination (PCV-13 + PPV 23) and 2020–2021 Seasonal Influenza vaccination in patients with comorbidities in France in 2020 (multivariable results).**
(DOCX)

**S1a Fig. Pneumococcal vaccine coverage rate in France in 2020 in at risk patients living with comorbidities (primary vaccination schedule - PCV13＋PPSV23).**
(DOCX)

**S1b Fig. Pneumococcal vaccine coverage rate in France in 2020 in immunocompromised patients (primary vaccination schedule - PCV13＋PPSV23).**
(DOCX)

**S2a Fig. Influenza vaccine coverage rate in France in season 2020–2021 in at risk patients living with an immunocompromised status.**
(DOCX)

**S2b Fig. Influenza vaccine coverage rate in France in 2020–2021 for immunocompromised patients.**
(DOCX)

**S1 File. Vaccine coverage rate per region.**
(DOCX)

## Acknowledgments

We thank the *Direction de la Stratégie, des études et des Statistiques*, *Département Accès, Traitements et Analyse de la Donnée*, and *Cellule de la CNAM en Charge de l'accompagnement des Demandes D'extraction* teams at the *Caisse Nationale de l'Assurance maladie* for data extraction.

The authors gratefully acknowledge the contributions of Gwendoline Poinsot (ORCID 0000-0002-5493-3506), Baptiste Pitel (ORCID 0000-0001-6007-2071), Marine Ginoux (ORCID 0000-0001-6441-5617) for statistical analyses, and data management supervision, respectively.

## Author contributions

**Conceptualization:** Benjamin Wyplosz, Benjamin Grenier, Nicolas Roche, François Roubille, Paul Loubet, Ariane Sultan, Bertrand Fougère, Jérôme Fernandes, Didier Duhot, Bruno Moulin, Fanny Raguideau, Emmanuelle Blanc, Gwenael Goussiaume.

**Formal analysis:** Benjamin Wyplosz, Benjamin Grenier, Nicolas Roche, François Roubille, Paul Loubet, Ariane Sultan, Bertrand Fougère, Jérôme Fernandes, Didier Duhot, Bruno Moulin, Fanny Raguideau, Emmanuelle Blanc, Gwenael Goussiaume.

**Investigation:** Benjamin Wyplosz, Benjamin Grenier, Nicolas Roche, François Roubille, Paul Loubet, Ariane Sultan, Bertrand Fougère, Jérôme Fernandes, Didier Duhot, Bruno Moulin, Fanny Raguideau, Emmanuelle Blanc, Gwenael Goussiaume.

**Methodology:** Benjamin Wyplosz, Benjamin Grenier, Nicolas Roche, François Roubille, Paul Loubet, Ariane Sultan, Bertrand Fougère, Jérôme Fernandes, Didier Duhot, Bruno Moulin, Fanny Raguideau, Emmanuelle Blanc, Gwenael Goussiaume.

**Writing – original draft:** Benjamin Wyplosz, Benjamin Grenier, Fanny Raguideau, Gwenael Goussiaume.

**Writing – review & editing:** Benjamin Wyplosz, Benjamin Grenier, Nicolas Roche, François Roubille, Paul Loubet, Ariane Sultan, Bertrand Fougère, Jérôme Fernandes, Didier Duhot, Bruno Moulin, Fanny Raguideau, Emmanuelle Blanc, Gwenael Goussiaume.

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
