## [Decision Letter · Decision Letter 0]

17 Mar 2025

PONE-D-24-58613Will vaccination of adults aged 65 years help to increase pneumococcal vaccination coverage in the at-risk population? A review of the evidence from a retrospective population-based study in France.PLOS ONE?

Dear Dr. Wyplosz,

Thank you for submitting your manuscript to PLOS ONE. After careful consideration, we feel that it has merit but does not fully meet PLOS ONE’s publication criteria as it currently stands. Therefore, we invite you to submit a revised version of the manuscript that addresses the points raised during the review process.

We look forward to receiving your revised manuscript.

Kind regards,

David J. Diemert, M.D.

Academic Editor

PLOS ONE

**Journal Requirements:**

1. When submitting your revision, we need you to address these additional requirements. Please ensure that your manuscript meets PLOS ONE's style requirements, including those for file naming. The PLOS ONE style templates can be found at https://journals.plos.org/plosone/s/file?id=wjVg/PLOSOne_formatting_sample_main_body.pdf and https://journals.plos.org/plosone/s/file?id=ba62/PLOSOne_formatting_sample_title_authors_affiliations.pdf 2. Thank you for stating the following financial disclosure: The study was funded by Pfizer Vaccines France.Please state what role the funders took in the study.  If the funders had no role, please state: "The funders had no role in study design, data collection and analysis, decision to publish, or preparation of the manuscript." If this statement is not correct you must amend it as needed. Please include this amended Role of Funder statement in your cover letter; we will change the online submission form on your behalf. 3. Thank you for stating the following in the Competing Interests section: [I have read the journal's policy and the authors of this manuscript have the following competing interests: BW received Payments from for lectures from Pfizer, Sanofi, Lilly, GSK, for meetings from Pfizer. BG and FR are employees of Heva, the CRO which received payments from Pfizer to conduct the study. NR received institutional grants, consulting fees, honoraria for lectures from Boehringer Ingelheim, Novartis, GSK, Pfizer, consulting fees and honoraria for lectures from AstraZeneca, Sanofi, Chiesi, Novartis, Teva, consulting fees from Bayer, Autral and Biosency, honoraria for lectures, from Zambon, MSD, Menarini support for meetings from Chiesi, AstraZeneca, GSK. FRoub received honoraria for lectures from Astra Zeneca, Servier, Boehringer, Astra Zeneca, Vifor, Bayer, Pfizer, Novartis, Servier, Novonordisk, Air liquid, Abbott, QuidelOrtho, Newcard, MSD, BMS, Sanofi, Alnylam, Zoll, Implicity, GSK, BMS, consulting fees from Abbott, Air liquide Bayer, Pfizer, support for travel from Novartis, Boehringer-Ingelheim, participates in an advisory board for Carmat, fiduciary role for Boehringer-Ingelheim, Vifor Pharma, Novartis, medical writting from Pfizer. PL received consulting fees from Pfizer, GSK, Sanofi, honoraria for lecture from Pfizer, GSK, Sanofi, Moderna, MSD, Support for attending meeting from Pfizer, Astrazeneca, MSD, Sanofi, GSK. AS received Grants from VIATRIS, SERVIER, Consulting fees from PFIZER, SANOFI, LILLY, SERVIER, ASTRA, DEBEX, NOVARTIS, URGO, honoraria for lectures from PFIZER, SANOFI, LILLY, SERVIER, ASTRA, DEBEX, NOVARTIS, ABOTT, MSD. BF received Consulting fees from Sanofi, AstraZeneca, Pfizer, GSK, honoraria for lectures from Sanofi, AstraZeneca, Pfizer, GSK, CSL Seqirus, Participation on a Data Safety Monitoring Board or Advisory Board for Pfizer. JF received Grants from Astra ZENECA ASTELLAS Pharma ABBVIE JANSSEN BMS, Consulting fees from Astra ZENECA ASTELLAS Pharma ABBVIE JANSSEN BMS, honoraria for lectures from Astra ZENECA ASTELLAS Pharma ABBVIE JANSSEN BMS. DD received honoraria for lecture from Pfizer, BM received honoraria for lecture from Pfizer. EB and GG are employees from Pfizer and may hold stock optionsWe note that one or more of the authors are employed by a commercial company. a. Please provide an amended Funding Statement declaring this commercial affiliation, as well as a statement regarding the Role of Funders in your study. If the funding organization did not play a role in the study design, data collection and analysis, decision to publish, or preparation of the manuscript and only provided financial support in the form of authors' salaries and/or research materials, please review your statements relating to the author contributions, and ensure you have specifically and accurately indicated the role(s) that these authors had in your study. You can update author roles in the Author Contributions section of the online submission form. Please also include the following statement within your amended Funding Statement. “The funder provided support in the form of salaries for authors [insert relevant initials], but did not have any additional role in the study design, data collection and analysis, decision to publish, or preparation of the manuscript. The specific roles of these authors are articulated in the ‘author contributions’ section.”If your commercial affiliation did play a role in your study, please state and explain this role within your updated Funding Statement.  b. Please also provide an updated Competing Interests Statement declaring this commercial affiliation along with any other relevant declarations relating to employment, consultancy, patents, products in development, or marketed products, etc.   Within your Competing Interests Statement, please confirm that this commercial affiliation does not alter your adherence to all PLOS ONE policies on sharing data and materials by including the following statement: "This does not alter our adherence to  PLOS ONE policies on sharing data and materials.” (as detailed online in our guide for authors http://journals.plos.org/plosone/s/competing-interests) . If this adherence statement is not accurate and  there are restrictions on sharing of data and/or materials, please state these. Please note that we cannot proceed with consideration of your article until this information has been declared. Please include both an updated Funding Statement and Competing Interests Statement in your cover letter. We will change the online submission form on your behalf. 4. Please amend either the title on the online submission form (via Edit Submission) or the title in the manuscript so that they are identical.

Reviewers' comments:

Reviewer's Responses to Questions

**Comments to the Author**

1. Is the manuscript technically sound, and do the data support the conclusions?

Reviewer #1: Yes

Reviewer #2: Yes

2. Has the statistical analysis been performed appropriately and rigorously?

Reviewer #1: Yes

Reviewer #2: Yes

3. Have the authors made all data underlying the findings in their manuscript fully available?

Reviewer #1: Yes

Reviewer #2: Yes

4. Is the manuscript presented in an intelligible fashion and written in standard English?

Reviewer #1: Yes

Reviewer #2: Yes

**Reviewer #1:**  Congratulations on a well crafted article that addresses a very important issue.

The article is well written, technical sound and addresses the objectives of the study. The only revision I would recommend is that of the title which is a bit 'wordy' and ambigous.

**Reviewer #2:**  The manuscript explores the factors associated with pneumococcal vaccination among older adults in France using population-level data. Overall, the study is well-structured and uses robust data sources and statistical methods. However, I recommend conducting additional sensitivity analyses and addressing several minor errors to enhance the manuscript.

Major Revisions:

Methods:

1. On Page 7, line 134: Please clarify whether the definition of vaccination coverage includes participants who are only partially vaccinated, such as those who received one dose of PCV13 or PPV23, or those who received two doses but did not follow the recommended schedule. If these participants are not included in the current definition, conducting a sensitivity analysis to explore their impact would be useful.

Results:

1. I recommend providing more interpretable explanations of the Odds Ratios (OR) alongside their categorization as negative, small, medium, or large effects. For instance, on line 265, the OR for "2 comorbidities" can be presented as: Individuals with two comorbidities are 2.26 times [95% CI] more likely to be vaccinated than those with one comorbidity, after adjusting for other specified variables. Furthermore, the effect is even more pronounced (OR = 3.75 [95% CI]) for those with three or more comorbidities compared to those with one comorbidity. You don't need to interpret every OR, but offering a clear interpretation for at least one OR in each effect category would help guide the audience through the results section.

Minor Revisions:

1. Page 3, line 33: Please specify the validated algorithms used in your analysis.

2. Page 4, line 58: I suggest incorporating data from the following reference for additional background on the global and France-specific incidence of invasive pneumococcal disease (IPD) by age: DOI:10.1016/S1473-3099(24)00665-0. Also, on line 67, remove the period before the reference.

3. Page 5, line 79: Spell out "EEA" when first mentioned. Additionally, on line 86, please specify the denominator for the statistic mentioned; consider adding the total number of adults in the French population. On line 90, spell out "GP."

4. Page 8, lines 150-156: The rationale for using OR in a large population-level study may be evident to readers with an epidemiology background. Consider moving this justification to the supplementary materials if you feel it is necessary to include.

5. Page 14, line 264: Correct "CI95%" to "95% CI."

6. Address any additional grammar and formatting issues, which may be better handled by the journal's editorial team during the final review process.

**Do you want your identity to be public for this peer review?** For information about this choice, including consent withdrawal, please see our Privacy Policy

Reviewer #1: No

Reviewer #2: No

---

## [Author Response · Author response to Decision Letter 1]

1 Jul 2025

Dear Dr Diemert,

We would like to thank both reviewers for their careful reading of our manuscript and their thoughtful comments of our submission PONE-D-24-58613 entitled:

“Will vaccination of adults aged 65 years help to increase pneumococcal vaccination coverage in the at-risk population? A review of the evidence from a retrospective population-based study in France.”

We answered to all the points mentioned as listed below.

We look forward to your consideration and response.

Yours sincerely,

Dr Benjamin Wyplosz

Journal requirements

1. When submitting your revision, we need you to address these additional requirements

3. Competing Interests

a. You can update author roles in the Author Contributions section of the online submission form

b. adherence statement

Updated Funding statement: “The financial support for this study was provided by Pfizer. The funders played no part in the study's design, data collection and analysis, or the decision to publish. GG and EB, who are employed by Pfizer, provided assistance with the preparation of the manuscript by offering critical feedback on the content prior to its submission.

The funder provided support in the form of salaries for authors BW, NR, FR, PL, AS, BF, JF, DD, BM, but did not have any additional role in the study design, data collection and analysis, or decision to publish. The specific roles of these authors are articulated in the ‘author contributions’ section.”

Updated Competing Interests Statement: “I have read the journal's policy and the authors of this manuscript have the following competing interests: BW received Payments from for lectures from Pfizer, Sanofi, Lilly, GSK, for meetings from Pfizer. BG and FR are employees of Heva, the CRO which received payments from Pfizer to conduct the study. NR received institutional grants, consulting fees, honoraria for lectures from Chiesi, GSK, Pfizer, consulting fees and honoraria for lectures from AstraZeneca, Sanofi, Teva, consulting fees from Austral and Biosency, honoraria for lectures, from Zambon, MSD, Menarini and support for meetings from Chiesi, AstraZeneca, GSK. FRoub received honoraria for lectures from Astra Zeneca, Servier, Boehringer, Astra Zeneca, Vifor, Bayer, Pfizer, Novartis, Servier, Novonordisk, Air liquid, Abbott, QuidelOrtho, Newcard, MSD, BMS, Sanofi, Alnylam, Zoll, Implicity, GSK, BMS, consulting fees from Abbott, Air liquide Bayer, Pfizer, support for travel from Novartis, Boehringer-Ingelheim, participates in an advisory board for Carmat, fiduciary role for Boehringer-Ingelheim, Vifor Pharma, Novartis, medical writting from Pfizer. PL received consulting fees from Pfizer, GSK, Sanofi, honoraria for lecture from Pfizer, GSK, Sanofi, Moderna, MSD, Support for attending meeting from Pfizer, Astrazeneca, MSD, Sanofi, GSK. AS received Grants from VIATRIS, SERVIER, Consulting fees from PFIZER, SANOFI, LILLY, SERVIER, ASTRA, DEBEX, NOVARTIS, URGO, honoraria for lectures from PFIZER, SANOFI, LILLY, SERVIER, ASTRA, DEBEX, NOVARTIS, ABOTT, MSD. BF received Consulting fees from Sanofi, AstraZeneca, Pfizer, GSK, honoraria for lectures from Sanofi, AstraZeneca, Pfizer, GSK, CSL Seqirus, Participation on a Data Safety Monitoring Board or Advisory Board for Pfizer. JF received Grants from Astra ZENECA ASTELLAS Pharma ABBVIE JANSSEN BMS, Consulting fees from Astra ZENECA ASTELLAS Pharma ABBVIE JANSSEN BMS, honoraria for lectures from Astra ZENECA ASTELLAS Pharma ABBVIE JANSSEN BMS. DD received honoraria for lecture from Pfizer, BM received honoraria for lecture from Pfizer. EB and GG are employees from Pfizer and may hold stock options. This does not alter our adherence to PLOS ONE policies on sharing data and materials.”

Reviewer #1

The only revision I would recommend is that of the title which is a bit 'wordy' and ambigous.

Answer: Thank you, we propose to simplify the title from “Will vaccination of adults aged 65 years help to increase pneumococcal vaccination coverage in the at-risk population? A retrospective population-based study in France” to “Pneumococcal vaccination at 65 years and vaccination coverage in at-risk adults: A retrospective population-based study in France.”

Reviewer #2

Methods: On Page 7, line 134: Please clarify whether the definition of vaccination coverage includes participants who are only partially vaccinated, such as those who received one dose of PCV13 or PPV23, or those who received two doses but did not follow the recommended schedule. If these participants are not included in the current definition, conducting a sensitivity analysis to explore their impact would be useful.

Answer: Thank you, we clarified pneumococcal vaccination coverage definition in the methods section: “corresponding to the reimbursement of one dose of PCV13 and one dose of PPV23 less than one year later”.

We added more analyses in Table 1 including subjects who received one dose of PCV13 or PPSV23, with the following text in the results section “More sensitive definitions of the vaccination coverage (at least one PCV13, at least onePCV13 and/or one PPSV23) reported higher rates, regardless of being immunocompromised or living with a comorbidity (Table 1)”. The significance of the corresponding increase in vaccination coverage rates is not easy to determine because they are based on incomplete or inaccurate schedules.

Results

Major revisions

1. I recommend providing more interpretable explanations of the Odds Ratios (OR) alongside their categorization as negative, small, medium, or large effects. For instance, on line 265, the OR for "2 comorbidities" can be presented as: Individuals with two comorbidities are 2.26 times [95% CI] more likely to be vaccinated than those with one comorbidity, after adjusting for other specified variables. Furthermore, the effect is even more pronounced (OR = 3.75 [95% CI]) for those with three or more comorbidities compared to those with one comorbidity. You don't need to interpret every OR, but offering a clear interpretation for at least one OR in each effect category would help guide the audience through the results section.

Thank you, we agree that interpreting briefly the OR at the beginning of each important section provides a more comprehensive lecture of our manuscript, which has been revised accordingly. Modifications were made from line 275 to line 317 of the manuscript (line numbers in the unmarked version without track change).

Minor revisions

1. Page 3, line 33: Please specify the validated algorithms used in your analysis.

Thank you, the algorithms are detailed in S1 methods, we modified the methods section by adding a complement to the following sentence: “Four algorithms were used by French Health authorities for the surveillance of HIV, end-stage renal failure with replacement therapy, chronic liver disease, and diabetes requiring treatment [24]; three were modified for our study (patients treated for cancer, auto-immune diseases, chronic respiratory diseases); the one for severe asthma was already published [25], and seven were created for our study (asplenia, hereditary immune deficit, solid organ transplantation, cyanotic cardiac disease, osteo-meningeal breach, and cochlear implant). We did not formally validate those latter, but we assessed the external validity by comparing the figures with available external data [26–28].”.

2. Page 4, line 58: I suggest incorporating data from the following reference for additional background on the global and France-specific incidence of invasive pneumococcal disease (IPD) by age: DOI:10.1016/S1473-3099(24)00665-0. Also, on line 67, remove the period before the reference.

Thank you for this very good reference. We incorporated relevant results from this publication, including on pneumococcal serotype replacement “However, the incidence of IPD is also a dynamic process that evolves in response to the use of pneumococcal conjugate vaccines. The PSERENADE (Pneumococcal Serotype Replacement and Distribution Estimation) project showed that 30 European countries saw a rapid reduction in the incidence of total IPD in young children, reaching 58-74% in the 6 years following the introduction of PCV10 or PCV13 into their routine vaccination schedules”.

3. Page 5, line 79: Spell out "EEA" when first mentioned. Additionally, on line 86, please specify the denominator for the statistic mentioned; consider adding the total number of adults in the French population. On line 90, spell out "GP."

Thank you. The abbreviation "EEA" is mentioned twice in the manuscript: first, it is spelled out at the beginning of the second paragraph of the introduction, and second, at the beginning of the third paragraph. We added the denominator “(N = 49,232,522)”. The GP abbreviation was spelled out on line 90.

4. Page 8, lines 150-156: The rationale for using OR in a large population-level study may be evident to readers with an epidemiology background. Consider moving this justification to the supplementary materials if you feel it is necessary to include.

Thank you, we edited this part of the methods (lines 169 to 174 in the unmarked version without track change).

5. Page 14, line 264: Correct "CI95%" to "95% CI."

Thank, you, we made the correction.

6. Address any additional grammar and formatting issues, which may be better handled by the journal's editorial team during the final review process.

Thank you, we had a naïve proofreading of all documents submitted.

---

## [Decision Letter · Decision Letter 1]

21 Jul 2025

Pneumococcal vaccination at 65 years and vaccination coverage in at-risk adults: A retrospective population-based study in France.

PONE-D-24-58613R1

Dear Dr. Wyplosz,

We’re pleased to inform you that your manuscript has been judged scientifically suitable for publication and will be formally accepted for publication once it meets all outstanding technical requirements.

Kind regards,

David J. Diemert, M.D.

Academic Editor

PLOS ONE

Additional Editor Comments (optional):

Reviewers' comments:

Reviewer's Responses to Questions

**Comments to the Author**

Reviewer #2: All comments have been addressed

2. Is the manuscript technically sound, and do the data support the conclusions?

Reviewer #2: Yes

3. Has the statistical analysis been performed appropriately and rigorously?

Reviewer #2: Yes

4. Have the authors made all data underlying the findings in their manuscript fully available?

Reviewer #2: No

5. Is the manuscript presented in an intelligible fashion and written in standard English?

Reviewer #2: Yes

Reviewer #2: The authors have thoroughly addressed the comments raised during the initial review, and the revised manuscript reflects significant improvements in clarity, methodological detail, and interpretability of the results. The addition of more interpretable interpretations of the odds ratios, clarification of vaccination coverage definitions, incorporation of relevant external references, and attention to minor formatting and editorial suggestions demonstrate the authors' responsiveness and commitment to improving the manuscript.

Notably, the updated analyses and clearer explanations provide compelling support for the authors' conclusion that an age-based recommendation for pneumococcal vaccination (as currently done for influenza in France) may enhance vaccine coverage among at-risk adults. The large, population-based dataset, along with the well-justified statistical approach, offers robust and policy-relevant findings.

Given that all major and minor concerns have been adequately addressed, I recommend acceptance of this revised version for publication.

**Do you want your identity to be public for this peer review?** For information about this choice, including consent withdrawal, please see our Privacy Policy

Reviewer #2: No

---

## [Editor Report · Acceptance letter]

PONE-D-24-58613R1

PLOS ONE

Dear Dr. Wyplosz,

I'm pleased to inform you that your manuscript has been deemed suitable for publication in PLOS ONE. Congratulations! Your manuscript is now being handed over to our production team.

Kind regards,

on behalf of

Dr. David J. Diemert

Academic Editor

PLOS ONE